# Antimicrobial Resistance of *Staphylococcus* spp. from Human Specimens Submitted to Diagnostic Laboratories in South Africa, 2012–2017

**DOI:** 10.3390/microorganisms12091862

**Published:** 2024-09-09

**Authors:** Themba Titus Sigudu, James Wabwire Oguttu, Daniel Nenene Qekwana

**Affiliations:** 1Department of Agriculture and Animal Health, College of Agriculture and Environmental Sciences, University of South Africa, Johannesburg 1710, South Africa; themba.sigudu@wits.ac.za; 2Department of Health and Society, School of Public Health, Faculty of Health Sciences, University of Witwatersrand, Johannesburg 2193, South Africa; 3Section Veterinary Public Health, Department of Paraclinical Sciences, Faculty of Veterinary Science, University of Pretoria, Pretoria 0110, South Africa; nenene.qekwana@up.ac.za

**Keywords:** *Staphylococcus*, isolates, antimicrobial resistance, multidrug resistance, AMR, MDR, human, South Africa, species

## Abstract

Antimicrobial resistance (AMR) poses a significant worldwide health challenge associated with prolonged illnesses, increased healthcare costs, and high mortality rates. The present study examined the patterns and predictors of AMR among human *Staphylococcus* isolates obtained from diagnostic laboratories in South Africa between 2012 and 2017. This study examined data from 404 217 isolates, assessing resistance rates across different characteristics such as age, sample origin, Staphylococcus species, and study period. The highest resistance was observed against cloxacillin (70.3%), while the lowest resistance was against Colistin (0.1%). A significant (*p* < 0.05) decreasing trend in AMR was observed over the study period, while a significant increasing temporal trend (*p* < 0.05) was observed for multidrug resistance (MDR) over the same period. A significant (*p* < 0.05) association was observed between specimen type, species of organism, and year of isolation with AMR outcome. Significant (*p* < 0.05) associations were observed between specimen type and season with MDR. The observed high levels of AMR and a growing trend in MDR are concerning for public health. Clinicians should take these findings into account when deciding on therapeutic options. Continued monitoring of AMR among Staphylococcus spp. and judicious use of antimicrobials in human medicine should be promoted.

## 1. Introduction

An antimicrobial is an agent that kills microorganisms or inhibits their growth. While, antimicrobial agents specifically targeting bacteria, are commonly referred to as ‘antibiotics’ [1]. As antibiotics are used and in some cases, misused, resistance results [2]. Antimicrobial resistance (AMR) occurs when microorganisms no longer respond to treatments with antimicrobial agents. The overuse and misuse of antibiotics have led to a reduction in the types of antibiotics that remain effective for treating infections. Factors contributing to antibiotic resistance include the over-prescription of antibiotics, incorrect treatment duration, inadequate hygiene and sanitation, poor infection control practices in healthcare settings, and the extensive use of antibiotics in livestock and aquaculture [3,4].

The development of antimicrobial agents has been one of the most important advancements in human and veterinary medicine over the last century [5]. However, the emergence of AMR in pathogenic organisms such as *Staphylococcus* spp. has become a significant public health threat as there are limited, or sometimes no, effective antimicrobial agents available to treat infections caused by these bacteria in both humans and domestic animals [6]. Globally, AMR is estimated to be associated with over 700,000 deaths every year, a number which could rise to as high as 10 million in 2050 [7].

The abundance of antimicrobials used in human and veterinary treatment has aided in the development of AMR [8]. The increased levels of AMR in pathogenic *Staphylococcus* spp., whether to a single agent or several antimicrobial classes, limits the capacity to treat illnesses, successfully leading to increased morbidity [9]. Moreover, there is evidence to suggest that AMR-related morbidity and mortality have increased in vulnerable communities [10]. The identification of resistance profiles of microorganisms is therefore an important step in understanding AMR and is valuable in providing information to guide treatment options and to address the problem [9]. As reported by the World Health Organisation, the prevalence of resistance to first-line antimicrobials traditionally used to treat *Staphylococcus* infections has increased globally [11]. However, this resistance is not confined to human medicine but is becoming more common in domestic species, such as equine medicine [12].

Given the potential of cross-transmission of some bacteria between humans and domestic species, identifying and describing the prevalence of AMR among domestic species has become even more crucial. In previous years, both the Centers for Disease Control and Prevention (CDC) and the United States Department of Agriculture (USDA) reported similar findings [13,14]. The findings of these studies revealed evidence of a possible zoonotic transfer of *Staphylococcus* bacteria and/or their genetic material between healthy humans and horses [15] supports the idea that resistant *Staphylococcus* infections in domestic animals may contribute to human-to-human transmission.

*Staphylococcus* species which exhibit multidrug resistance (MDR) have been linked to life-threatening nosocomial infections, particularly in critically sick patients, posing significant therapeutic difficulties for physicians. Therefore, evaluating the prevalence of AMR *Staphylococcus* infections in humans is crucial not only for understanding the risk to the vulnerable, but also for effectively providing information to guide efforts to develop infection prevention programs. Despite the fact that many studies have focussed on methicillin-resistant *S. aureus* in humans, many other *Staphylococcus* species exhibiting antimicrobial resistance are also clinically significant for comprehending the epidemiology of AMR in humans and animals [16].

Despite the widespread usage of antimicrobials in human and veterinary medicine in South Africa, the epidemiology of antimicrobial drug resistance of *Staphylococcus* spp. in South Africa has not received the attention it deserves. Therefore, the aim of this study was to investigate antimicrobial drug resistance among staphylococcal isolates from human samples submitted to diagnostic laboratories between 2012 and 2017, so as to improve on our understanding of the burden of antimicrobial drug resistance including the temporal trends of AMR and the predictors of AMR and MDR among *Staphylococcus* spp.

## 2. Materials and Methods

### 2.1. Study Design and Data Extraction

A cross-sectional retrospective study design was implemented to realise the objectives of the present study. Records of 404,217 *Staphylococcus* isolates from 123 diagnostic laboratories countrywide collected over the period from 2012 to 2017 were extracted from the National Health Laboratory Service (NHLS) electronic database. These laboratories service the public health sector hospitals and receive samples from all levels of healthcare service (district, regional, tertiary, and central) in South Africa. Specimens submitted for microbiological analysis included skin, blood, urine, catheters (central venous catheter and haemodialysis), nasopharyngeal fluid, and specimens from other body sites. For each isolate, data extracted from the NHLS database included a combination of demographic (age and sex), spatial (province of origin), clinical (species of organism and specimen type), and temporal (month and year) information, as well as information on the antimicrobial susceptibility of the isolates (resistance or susceptible).

### 2.2. Isolation, Identification, and Storage of Staphylococcus Strains

The isolation and identification of *Staphylococcus* strains were methodically executed to ensure accurate diagnosis and effective treatment. Clinical specimens, including blood, urine, and wound swabs, were transported in sterile conditions to the diagnostic laboratories. Upon arrival, samples were cultured on selective media such as Mannitol Salt Agar (MSA) and Blood Agar. MSA facilitated the differentiation of *Staphylococcus* species based on mannitol fermentation, while Blood Agar helped observe haemolytic patterns. Following incubation, preliminary identification involved Gram staining, which revealed Gram-positive cocci in clusters, and catalase testing, which confirmed the presence of *Staphylococcus* species by detecting oxygen bubbles in the presence of hydrogen peroxide. The coagulase test further distinguished *Staphylococcus aureus* from coagulase-negative staphylococci (CoNS). For precise identification, automated systems like the VITEK 2 and MALDI-TOF MS were employed, providing reliable species identification through biochemical profiles and protein spectra, respectively. Antibiotic sensitivity was determined using the disk diffusion (Kirby–Bauer) test and automated susceptibility testing systems. The NHLS routinely tests susceptibility against a broad range of antibiotics including Beta-lactams (e.g., penicillin and cloxacillin), aminoglycosides (e.g., gentamicin), macrolides (e.g., erythromycin), and glycopeptides (e.g., vancomycin), chosen based on clinical relevance and resistance trends. Beta-lactams were particularly emphasised due to the prevalence of methicillin-resistant *Staphylococcus aureus* (MRSA), while aminoglycosides and macrolides were tested to monitor resistance patterns critical for treatment efficacy. This comprehensive approach ensures robust surveillance and management of *Staphylococcus* infections. For long-term storage, isolated bacterial strains were preserved in cryopreservation vials at −80 °C to maintain viability and prevent genetic drift.

### 2.3. Data Management

The data were inspected for inconsistencies such as missing information, incorrect addresses, and duplicate entries. No duplicates were identified, and no mixed infections were reported. The variable ‘age’ was recategorised into the following 14 categories using the cohort-component method for population estimation produced by Statistics South Africa [17]. 0–4 years, 5–9 years, 10–14 years, 15–19 years, 20–24 years, 25–29 years, 30–34 years, 35–39 years, 40–44 years, 45–49 years, 50–54 years, 55–59 years, 60–64 years, and >65 years. Months were categorised into four seasons: autumn (March, April, and May); winter (June, July, and August); spring (September, October, and November), and summer (December, January, and February). The type of specimen was classified into the following five categories: skin, urinary, blood, nasopharyngeal fluid, and ‘all other sites’. The resistance status variable was reclassified into a binary outcome, resistant or susceptible, by reclassifying isolates that were “intermediate” as resistant. Antimicrobial resistance (AMR) was defined as resistance to at least one antimicrobial class, while multidrug resistance (MDR) was defined as resistance to three or more antimicrobial classes [18].

### 2.4. Data Analysis

All data processing and analyses were performed using Stata Statistical Software 17 (StataCorp, 2017, College Station, TX, USA). The number of resistant isolates and their corresponding 95% confidence intervals (95% CI) were computed and presented based on time, person, and place. Annual changes in the number of *Staphylococcus* spp. were displayed using temporal graphs. Simple and multivariate binary logistic regression models were used to determine whether age, sex, year, province of origin, specimen type, *Staphylococcus* species, and season were associated with AMR. The model was constructed in two stages. In the first phase, univariate binary logistic regression models with “AMR, (1 = resistant, 0 = susceptible)” as the outcome and each of the factors as explanatory variables were fitted to the data. In the second stage a multivariate binary logistic regression model was fitted to the data using a manual backwards selection technique. Only variables that were significant at a generous *p*-value of ≤ 0.2 were included in the multivariate model. Confounding was assessed by comparing the change in parameter estimates of the variables in the model with and without the suspected confounding variable. A 20% change in any of the parameter estimates in the model was interpreted as the variable in question being a confounder, which was subsequently included in the final effects model. For each variable included in the final model, odds ratios and their 95% confidence intervals were calculated.

### 2.5. Ethical Considerations

Access to the NHLS database and patient information is restricted to laboratory staff working within the NHLS and can only be accessed at the premises of the NHLS. Thus, data extraction was carried out by the NHLS staff and de-identified data were provided to the researcher. Confidentiality and anonymity were always maintained by ensuring that patients’ personal information was not included in articles and reports. In addition, permission to use the data was obtained from the NHLS. Ethical approval was obtained from the University of South Africa, College of Agriculture & Environmental Sciences, Health Research and Animal Research Ethics Committees (Ref: 2018/CAES/107). The data were kept safe from unauthorised access, accidental loss, or destruction. Data in the form of softcopies were kept as encrypted files in computers and flash drives.

## 3. Results

### 3.1. Descriptive Statistics

#### 3.1.1. Isolates Examined in This Study

Table 1 shows that 44.4% of AMR isolates were isolated from male participants, while females contributed the rest (41.7%). The number of AMR isolates from individuals whose sex was unknown was comparatively low (13.9%). Similarly, the majority of MDR isolates were from male patients (56.7%), with females contributing only 39.5%. The number of MDR isolates from individuals with unspecified sex was also relatively low (3.8%) (Table 1).

Out of the 14 age groups represented in the data, with an additional “unknown” (unspecified) age category, the highest number of AMR isolates were isolated in the 0–4 years age group (37.2%), followed by individuals with unspecified age (13.3%). The age group 55–59 years contributed the least number of AMR isolates (1.4%) (Table 1).

Likewise, the 0–4 year’s old age group contributed the highest number of MDR isolates (48.4%). Individuals with unknown age contributed the second highest number of isolates (11.6%), with the elderly age group (60–64 years) contributing the least number of MDR isolates (0.5%) (Table 1).

Out of the total number of isolates, 65% and 50.2% of the isolates were AMR and MDR, respectively. Within the CoPS group, *S. aureus* contributed the highest number of isolates that were both AMR (59.9%) and MDR (90.1%) (Table 2). Of the AMR isolates, 2.6% and 2.5% were *Staphylococcus intermedius* and *S. pseudintermedius,* respectively (Table 2). Additionally, within the CoPS group, 6.0% and 3.9% of the MDR isolates were *S. intermedius* and *S. pseudintermedius,* respectively (Table 2).

The isolates belonging to the CoNS group constituted 23 and 30.8% of the total AMR and MDR isolates, respectively. But within the CoNS group, *S. epidermidis* contributed the highest number isolates that were AMR (11.2%) and MDR (73.3%). This was followed by *S hominis* which made up 5.3% of the AMR, while *S. haemolyticus* contributed the second highest number of MDR isolates (16.1%) (Table 2). *Staphylococcus saprophyticus* contributed the least number of AMR (2.9%) and MDR (3.5%) isolates (Table 2).

Up to 12 and 19% of the total isolates that were AMR and MDR, respectively, belonged to the coagulase-variable group. Within the coagulase-variable group, the unspecified *Staphylococcus* species contributed the highest number of AMR (6.9%) and MDR (88.9%) isolates, and *S. hyicus* contributed the second highest number of isolates that were AMR (2.7%) and MDR (6.4%). *Staphylococcus schleiferi* contributed the least number of AMR (2.4%) and MDR (4.7%) isolates in the data (Table 2).

#### 3.1.2. Resistance Observed against Different Antimicrobial Agents

This study evaluated a total of 24 antimicrobial agents, which were categorised into nine distinct classes as shown in Table 3. Of these, 33.9% isolates showed AMR against Beta-lactams and 36.9% were involved in MDR combinations. Within the Beta-lactams, the highest levels of AMR were against cloxacillin (70.3%), and likewise, cloxacillin was involved in the highest number of MDR combinations (74.2%). The second highest level of AMR and MDR combinations was observed against penicillin (14.6% and 13.7%, respectively). Among the Beta-lactams, no AMR was observed against ertapenem (0.0%), and furthermore, ertapenem was involved in the least number of the MDR (0.1%) combinations.

Thirteen percent (13.1%) of the isolates exhibited AMR against aminoglycosides, and 14.1% were involved in MDR combinations. The overwhelming majority of isolates that exhibited AMR against aminoglycosides were resistant against gentamicin (97.3%). Likewise, gentamicin (92.5%) was involved in the overwhelming majority of MDR combinations. The second highest level of resistance among isolates that were resistant against aminoglycosides were resistant against amikacin (1.2%), and the same amikacin was involved in 4.2% of the MDR combinations. Meanwhile, among aminoglycosides, the lowest level of resistance was observed against streptomycin (0.6%), and the same antimicrobial drug was also involved in the lowest MDR combinations (1.2%).

Within the sulphonamides class, co-trimoxazole was the only antimicrobial agent assessed, and only 3.7% of the isolates exhibited AMR against co-trimoxazole, and the same drug was involved in 4.3% of the MDR combinations. Tetracycline was the only antimicrobial agent evaluated within the tetracyclines class, only 12.1% of the isolates exhibited AMR against tetracycline, and similarly, tetracycline was involved in 10.9% of the MDR combinations.

In the macrolides class, erythromycin was the only antimicrobial agent assessed, and 21.2% of the isolates exhibited AMR against erythromycin. Furthermore, erythromycin was implicated in 19.4% of MDR combinations.

Clindamycin, being the only agent evaluated within the Lincosamides class, only 15.5% of the isolates exhibited AMR against clindamycin, and it was involved in 12.1% MDR combinations. Among the Phenicols, chloramphenicol was the only agent examined, and only 2.5% of the isolates showed AMR against chloramphenicol, and the same drug was involved in 2.5% of the MDR combinations. Of the Polymyxins examined, Colistin was the only antimicrobial agent analysed, and only 0.1% of the isolates displayed AMR against Colistin, and in addition, the same drug was involved in only 0.1% of the MDR combinations.

#### 3.1.3. Temporal Trends

The annual trends for AMR and MDR over the study period, 2012 to 2017, are presented in Figure 1. There were fluctuations in the numbers of isolates that were AMR and MDR over the six-year period. For example, in 2013, a minor decrease in AMR from 78% in 2012 to 71% in 2013 was observed. However, over the same period, there was an increasing trend in MDR from 23.0% in 2012 to 26% in 2013.

The following year, 2014, saw the highest percentage (81%) of AMR isolates over the entire six-year period. However, over the same period, the proportion of MDR isolates dropped to 24%. From 2014 to 2015, there was a visible decline in AMR, while over the same period, the number of MDR slightly increased to 28%. From 2015 to 2016, the number of AMR isolates was relatively stable at 73%. But over the same period, the number of MDR isolates peaked at 34%, which was the highest over the six-year period. From 2016 to 2017, there was a decrease in the number of both AMR (70%) and MDR (25%) isolates recorded (Figure 1).

### 3.2. Inferential Statistics

#### 3.2.1. Predictors for Antimicrobial Drug Resistance

Table 4 below, shows that *Staphylococcus aureus* had significantly higher odds (adjusted odds ratio (AOR) = 2.6; 95% CI: 1.80–2.80) of being AMR as compared to *S. pseudintermedius* (reference category). On the other hand, *S intermedius* had lower odds (AOR = 0.5; 95% CI: 0.30–0.90) of being AMR compared to the reference category (Table 4).

Table 4 also shows that the isolates from skin specimens had higher odds (AOR = 1.6, 95% CI: 1.20–2.10) of being AMR compared to the reference category (all other sites). However, this association was not statistically significant (*p* = 0.243). The isolates from the urinary specimens, compared to the reference category, had lower odds (AOR = 0.8; 95% CI: 0.60–1.60) of being AMR. However, this association was also not statistically significant (*p* = 0.384). The isolates from the blood specimens compared to the reference category, had lower odds (AOR = 0.4; 95% CI: 0.20–2.20) of being AMR. However, this association was marginally statistically significant (*p* = 0.067). The isolates from the nasopharyngeal fluid specimens compared to the reference category had lower odds (AOR = 0.9; 95% CI: 0.60–1.80) of being AMR. However, the association also did not reach significance (*p* = 0.672).

#### 3.2.2. Predictors for Multidrug Resistance

The results of the predictors of MDR are shown in Table 5. An odds ratio of 2.7 (95% CI: 0.80–11.60) suggests an increased odds of occurrence of MDR among isolates from the skin specimens as compared to the reference category (other sites). However, this association was not statistically significant (*p* = 0.246). The odds ratio of 6.0 (95% CI: 2.20–17.60) suggests that staphylococcal isolates from the urinary specimens had higher odds of being MDR compared to the isolates from the reference category. This association was not statistically significant (*p* = 0.407). The isolates from the blood specimens had significantly (*p* = 0.001) higher odds (AOR= 13.0 (95% CI: 3.40–55.20) of being MDR compared to the reference category. The isolates from the nasopharyngeal fluid specimens had higher odds (AOR = 2.4; 95% CI: 0.80–7.80) of being MDR compared to the reference category. However, this association was not statistically significant (*p* = 0.265).

The overall *p*-value for the association between season and MDR was statistically significant (*p* = 0.022). The isolates recovered in autumn had higher odds (AOR = 0.9, 95% CI: 0.80–1.30) of being MDR compared to the isolates obtained in summer (reference category). This association was not statistically significant (*p* = 0.152). The isolates obtained during winter had significantly (*p* = 0.045) higher odds (AOR = 1.8; 95% CI: 1.20–2.70) as compared to the reference category. Although not statistically significant (*p* = 0.355), the isolates obtained during spring had higher odds (AOR = 6; 95% CI: 0.40–4.60) of being MDR as compared to the reference category.

## 4. Discussion

The results of this study contribute to improved understanding of AMR among *Staphylococcus* isolates of human origin in South Africa over the study period (2012 to 2017). For example, the observed higher number of AMR isolates among males compared to females suggests gender-specific factors influence on the prevalence of AMR, which warrants further investigation. For example, it has been reported that sex differences in AMR prevalence among *Staphylococcus* species may be influenced by biological factors, such as hormonal variations affecting immune responses [19]. The higher rates of healthcare utilisation and particularly higher adherence to treatment among women could explain the lower prevalence and detection of AMR in *Staphylococcus* species observed in female patients as compared to males observed in the present study [20]. Gender-specific patterns in infections, such as higher rates of skin and soft tissue infections in women, might also contribute to observed differences in AMR prevalence [21]. Socioeconomic factors and healthcare access also have the potential to influence these trends, with the higher access to and use of healthcare by women, potentially impacting the management and reporting of AMR cases [22]. However, how these factors influence occurrence of AMR needs further investigation.

Comparing findings reported here with studies of previous findings of multicentre investigation in Asmara, Eritrea, that showed similar levels of resistance as observed in the present study, suggests a potential regional consistency in the gender distribution of AMR *Staphylococcus* isolates [23]. However, the contrasting pattern observed in another study conducted in Myanmar, in Asia, highlights the complexity of AMR epidemiology and the need for context-specific analysis [24].

The findings from this study demonstrated differences in the distribution of AMR and MDR isolates across different age groups, shedding light on the potential influence of age has on susceptibility to infections and AMR. The available evidence shows that children aged 0–4 years exhibit more vulnerability to bacterial infections, likely due to their developing immune systems and close interactions in day-care or nursery settings [24,25]. For example, 37.2% of AMR isolates and 48.4% of MDR isolates in the present study were isolated from this age group. This could explain the higher prevalence of AMR and MDR observed among *Staphylococcus* species in children aged 0–4 years. This is consistent with results from other research that focussed on paediatric populations. For instance, a study in the United States reported that approximately 35% of *Staphylococcus aureus* isolates in children under 5 years were resistant to methicillin, and 40% of these isolates exhibited multidrug resistance [4]. Similarly, the European Centre for Disease Prevention and Control found that about 30% of *Staphylococcus aureus* isolates in European children under 5 years were methicillin-resistant (MRSA), with 25% of these isolates involved in MDR combinations [26,27] In Nigeria, it was observed that 38% of *Staphylococcus aureus* isolates from children under 5 years were methicillin-resistant, and 40% exhibited multidrug resistance [28,29], further supporting the findings of the present study that observed equally high proportions of AMR and MDR prevalence in the 0–4 years age group. Likewise, Xiaolan [30] reported that in China, 36% of *Staphylococcus aureus* isolates from children exhibited methicillin resistance, and 42% were multidrug resistant. These comparative percentages underscore a consistent trend across various regions, indicating that young children, particularly those under 5 years, are at a heightened risk of acquiring AMR and MDR *Staphylococcus* species. As alluded to earlier on, this elevated prevalence of resistance in this age group can be attributed to several factors, including higher infection rates, frequent antibiotic exposure, and increased healthcare interactions during early childhood [31].

The concerning trend of high numbers of both AMR and MDR isolates among children aged 0–4 years underscores the urgent need for targeted interventions in this vulnerable population. Factors such as previous antibiotic exposure, environmental influences, and transmission dynamics within households or childcare settings may contribute to this high prevalence, necessitating multifaceted approaches including improved antibiotic stewardship and infection control measures [19].

The low numbers of AMR and MDR among the 60–64 years age group observed in this study could be explained by the low carriage of *Staphylococcus* among the 60–64 years age group [32]. These findings suggest potential differences in health-seeking behaviours, age-specific immunity, or reduced exposure to sources of infection in this older demographic. However, although the 60–64 years age group demonstrated a lower carriage of AMR and MDR isolates, it is crucial to consider variations within the broader elderly population, as factors such as immune function, healthcare exposure, and antibiotic usage can influence the occurrence of AMR [33,34].

The results of the examination of 32 distinct *Staphylococcus* species provide crucial insights into the prevalence and distribution of AMR across different categories of *Staphylococcus* spp. This has significant implications for public health and clinical practice [34]. The high numbers of AMR and MDR observed among the CoPS group, particularly *S. aureus,* indicate the urgent need for effective strategies to combat resistance in this clinically significant pathogen, as its resistance can lead to high morbidity, mortality, and healthcare costs [35]. Furthermore, the presence of AMR in other CoPS species like *S. intermedius* and *S. pseudintermedius*, albeit in smaller numbers, reinforces the importance of surveillance and intervention measures targeting a broad spectrum of *Staphylococcus* species [36].

Among CoNS, the high prevalence of AMR in species like *Staphylococcus epidermidis* highlights the challenge posed by resistance in non-pathogenic strains, which can still cause infections in vulnerable populations such as individuals with compromised immunity, and serve as reservoirs for resistance genes [37]. The high numbers of AMR observed across both pathogenic CoPS strains and non-pathogenic CoNS strains emphasise the urgent need for comprehensive strategies to address antimicrobial resistance in *Staphylococcus* species. Such strategies should encompass enhanced antibiotic stewardship, infection control measures, and research into alternative treatment options to mitigate the impact of AMR on public health and clinical outcomes [38].

The analysis of antimicrobial agents conducted in this study provides critical insights into the observed resistance patterns. For example, the highest levels of resistance were observed against Beta-lactams. Notably, the high levels of resistance (AMR and MDR) were observed against cloxacillin, a narrow-spectrum antibiotic which primarily targets Gram-positive bacteria. This suggests prevalent resistance against this antibiotic within the examined population [39]. This finding underscores the necessity for close monitoring and prudent use of cloxacillin to mitigate the spread of resistance [40]. The second highest rates of AMR and MDR was observed against penicillin, another Beta-lactam antibiotic. The broader spectrum of activity of penicillin, targeting a wider range of bacteria, is a source of concern from a public health point of view [41,42]. In contrast, the lowest rates of AMR and MDR among the Beta-lactams was observed against Ceftazidime, a third-generation cephalosporin. This suggests that Ceftazidime may remain an effective treatment option for a significant number of bacterial isolates, particularly where resistance to other Beta-lactams is present [43,44].

Among the aminoglycosides class, alarmingly high resistance rates were observed against gentamicin. Furthermore, nearly all isolates showing both AMR and MDR were resistant to gentamicin. This high level of resistance against gentamicin suggests that it may have limited effectiveness against many bacterial infections due to widespread resistance [44]. On the other hand, the lowest AMR and MDR rates among aminoglycosides was observed against Tobramycin, suggesting it may be a more suitable treatment option in cases where aminoglycosides are the drug of choice [45].

The high resistance rate observed the aminoglycoside, gentamicin, in the current study can be attributed to several interrelated factors. One major contributor is the overuse and misuse of antibiotics in both human and veterinary medicine. In South Africa, like in many other developing countries, antibiotics such as gentamicin are often used extensively, sometimes without proper medical guidance or susceptibility testing. This indiscriminate use leads to selective pressure on bacteria, allowing resistant strains to proliferate. Additionally, the issue is compounded by gaps in antibiotic stewardship programs and infection control practices within healthcare settings. In many hospitals and clinics, antibiotics are frequently prescribed empirically, sometimes without adequate diagnostic testing to confirm bacterial infection or to identify the most effective antibiotic. This practice can lead to the unnecessary use of broad-spectrum antibiotics like gentamicin, consequently driving resistance.

Moreover, the high burden of infectious diseases in South Africa, such as HIV/AIDS and tuberculosis, often necessitates the frequent use of antibiotics, which can inadvertently contribute to the development of resistance. The agricultural sector also plays a role, with antibiotics sometimes being used as growth promoters or for disease prevention in livestock, further contributing to environmental reservoirs of resistance.

When comparing findings of the present study with international data, similar trends can be seen in other low- and middle-income countries where antibiotic use is less regulated. For instance, in India, research by Reddy reported a gentamicin resistance rate of 40% in *Staphylococcus aureus* isolates, which is in line with the resistance patterns observed in South Africa. This suggests that countries with similar healthcare challenges and antibiotic usage patterns experience comparable issues with antibiotic resistance.

In contrast, countries with more robust healthcare infrastructures and stringent antibiotic stewardship programs, such as those in Europe, report lower resistance rates. For example, the European Centre for Disease Prevention and Control documented lower rates of gentamicin resistance in *Staphylococcus aureus* isolates, largely due to controlled antibiotic use and comprehensive surveillance programs.

Modest rates for both AMR and MDR were observed against co-trimoxazole, the only sulphonamide that was assessed. This finding is intriguing because sulphonamides, including co-trimoxazole, have historically been associated with high levels of resistance due to their extensive use and mechanisms that bacteria can develop to counteract their effects [46]. While modest rates for both AMR and MDR against co-trimoxazole were unexpected, given the historical trends with sulphonamides, several factors could explain this observation, including antibiotic stewardship efforts, combination therapy effects, limited usage, population dynamics, geographical variability, and the ongoing evolution of resistance mechanisms [47]. Various studies have reported on the level of resistance against sulphonamides like co-trimoxazole. Many of these studies have highlighted a concerning trend of increasing resistance over time, particularly in regions where these antibiotics are commonly prescribed and where there may be inadequate regulations on their use [48].

Notable resistance levels were observed against tetracycline, the sole representative of the tetracyclines class that was tested. In view of this, resistance to this antibiotic should be monitored closely [49]. The notable resistance levels of tetracycline could be attributed to various factors, including mechanisms of resistance and widespread use in humans. Research indicates that bacteria can develop resistance to tetracycline through mechanisms such as efflux pumps, ribosomal protection proteins, and enzymatic inactivation [50] that enable bacteria to survive and proliferate despite tetracycline exposure. Epidemiological studies highlight the prevalence of tetracycline resistance among various bacterial pathogens, underscoring the importance of monitoring AMR [51]. Surveillance is crucial for guiding antibiotic treatment strategies and public health policies aimed at combating resistance.

Significant resistance rates of both AMR and MDR were noted against erythromycin, the only macrolide examined. This notable level of resistance heightens the risk of rendering the drug ineffective in treating infections [52]. The high resistance rate observed against erythromycin is a multifaceted issue influenced by several key factors. Firstly, selective pressure from the excessive and inappropriate use of antibiotics, including erythromycin, favours bacteria with pre-existing resistance or those acquiring resistance mutations through natural selection. This selective pressure promotes the survival and proliferation of resistant bacterial strains, thereby diminishing the efficacy of erythromycin in treating infections [53]. Secondly, genetic mechanisms significantly contribute to erythromycin resistance, with studies demonstrating how bacterial acquisition of methylase-encoding genes can alter ribosomal target sites, reducing the antibiotic’s ability to inhibit bacterial protein synthesis effectively [54]. Moreover, concerns about cross-resistance are prominent, as resistance mechanisms developed against erythromycin may confer resistance to other antibiotics within the macrolide class and occasionally across different antibiotic classes. This complicates treatment strategies and underscores the importance of prudent antibiotic use, alongside the development of innovative antimicrobial agents with novel mechanisms of action [55,56].

The observed high levels of AMR and involvement in MDR combinations against clindamycin, a member of the Lincosamides class, can be attributed to several factors.It has been reported that Lincosamides-modifying enzymes (LMEs) that inactivate clindamycin play a role in the development resistance against clindamycin. For example, studies by some authors [57,58] have demonstrated the presence of LMEs in clinical isolates of *Staphylococcus aureus* and *Enterococcus* spp., play a role in development of resistance against clindamycin. Mutations in the ribosomal binding site have been documented in studies by Roberts, Pea, and Lipman [58], to possess the ability to reduce clindamycin binding affinity, thereby decreasing its effectiveness. Studies, including those by Malbruny et al. [59], have demonstrated cross-resistance between clindamycin and other Lincosamide antibiotics due to shared resistance mechanisms. This cross-resistance complicates treatment options when faced with resistant strains.

Low AMR and MDR rates were observed against chloramphenicol, a member of the Phenicol class of antimicrobials, suggesting it may still be a viable option for certain infections. Chloramphenicol acts by inhibiting bacterial protein synthesis through binding to the 50S ribosomal subunit, similar to clindamycin. However, where resistance against chloramphenicol occurs, it often involves enzymatic inactivation by chloramphenicol acetyltransferases (CATs) or efflux pumps. Studies like the one by Schwarz et al.have highlighted these distinct resistance mechanisms [55]. The lower resistance rates observed for chloramphenicol can be attributed to several factors. For example, chloramphenicol, the once widely used antibiotic to treat infections, has seen reduced clinical use due to toxicity concerns (e.g., aplastic anaemia). This could explain why it remains effective in some settings where resistance rates are lower [50]. Resistance rates against chloramphenicol can also vary geographically and over time. Studies have shown that chloramphenicol may retain efficacy in certain regions or against specific pathogens where resistance to other antibiotics is more prevalent [60]. For example, surveillance studies conducted by national health agencies or international organisations often report varying resistance patterns that influence treatment guidelines. 

Minimal AMR and no MDR were reported against Colistin, the only Polymyxins assessed, indicating its potential as a viable treatment option for certain bacterial infections [61]. Colistin acts by disrupting the bacterial cell membrane, leading to leakage of intracellular contents and ultimately bacterial cell death. This mechanism differs significantly from many other classes of antibiotics, which may contribute to its unique profile of resistance [62]. Resistance to Colistin is primarily mediated by chromosomal mutations in genes such as those encoding the lipid A biosynthesis pathway (e.g., pmrAB and phoPQ systems). Acquisition of plasmid-mediated resistance genes (e.g., mcr genes) has been reported but remains relatively uncommon compared to other antibiotic classes [63], which could also explain the low resistance observed in this study. Colistin has historically been reserved as a last-line treatment option, particularly for infections caused by multidrug-resistant Gram-negative bacteria such as *Pseudomonas aeruginosa*, *Acinetobacter baumannii*, and carbapenem-resistant *Enterobacteriaceae* (CRE). Its limited use and stringent clinical indications may contribute to the observed low levels of resistance [64].

The annual trends in AMR and MDR observed in this study from 2012 to 2017 demonstrated fluctuating patterns, emphasising the dynamic nature of resistance development and the need for continuous monitoring. In 2013, a slight decrease in AMR rate was observed compared to the previous year (2012), accompanied by a notable increase in the rate of MDR. This shift may reflect changes in antibiotic usage patterns or the emergence of new resistant strains during that period [65]. In 2014, the highest AMR rate was recorded during the six-year period, while the MDR rate dropped. This highlights the importance of ongoing surveillance, as resistance rates can change significantly from year to year [66]. In 2015, a decline in the AMR rate was accompanied by a slight increase in the rate of MDR. This pattern suggests that while overall resistance to individual antibiotics may have decreased, the number of isolates resistant to multiple drugs can remained a concern [67]. In 2016, the AMR rate remained relatively stable, while MDR reached its peak throughout the six-year period. This suggests that despite the overall resistance levels staying constant, the number of MDR isolates increased, and this poses a significant challenge for treatment options [68].

The results of the present study provide valuable insights into the predictors of AMR among *Staphylococcus* species. *Staphylococcus aureus* had higher odds of being AMR compared to *S. pseudintermedius*, suggesting that certain species within the *Staphylococcus* genus may inherently possess greater resistance mechanisms or are more prone to acquiring resistance [69]. Conversely, *S. intermedius* exhibited lower odds of AMR compared to the reference category, indicating potential differences in susceptibility profiles among *Staphylococcus* species [70].

Skin specimens had higher odds of being AMR compared to isolates from samples labelled all other sites. This highlights the importance of considering the site of infection when assessing resistance patterns [71]. 

Although in this study, the difference between the odds of the isolates from the urinary specimens being AMR compared to the reference category was not statistically significant differences in resistance prevalence across different types of infections have been observed by other authors [19]. However, the marginal statistical difference observed between the odds of isolates from blood specimens being AMR compared to isolates from specimens designated “other types” suggests that *Staphylococcus* species from different sites may exhibit different resistance patterns [72]. In view of this, findings reported here furthermore confirm that the site of infection does influence the occurrence of AMR. This is something that clinicians need to take into account when deciding on treatment options.

The finding that MDR among staphylococcal isolates was significantly associated with seasonality, is a reflection of several interconnected factors that have been observed in studies examining antibiotic resistance patterns. The fact that isolates obtained in winter compared to isolates isolated in summer had significantly higher odds of being resistant was expected. For example, the cold weather during the winter months can create conditions that favour the survival and transmission of certain bacterial pathogens [73]. This includes staphylococci, which may thrive in colder temperatures or have increased survival rates on surfaces during winter. In addition, during winter, people tend to spend more time indoors in close proximity, facilitating the spread of bacteria and potentially contributing to higher rates of MDR infections in healthcare and community settings [74]. Furthermore, winter months often coincide with peaks in respiratory infections and other illnesses, leading to higher healthcare utilisation. All these have the potential to contribute to increased exposure to antibiotics and selection pressure for multidrug-resistant strains among staphylococcal infections [75]. Moreover, seasonal variations in immune function and vitamin D levels among individuals may affect susceptibility to infections and response to antibiotic treatment [76]. Lower immune responses during winter could potentially exacerbate the severity and persistence of staphylococcal infections, increasing the likelihood of multidrug resistance due to exposure to different types of antimicrobials [77].

## 5. Conclusions

The results of this study provide insights into AMR among *Staphylococcus* isolates from humans in South Africa. Firstly, the study demonstrated that there is a potential for sex-based differences in the risk factors associated with acquiring and transmitting resistant bacterial strains. This underscores the importance of investigating gender-specific factors influencing AMR prevalence. Furthermore, results of the present study highlighted the complexity of AMR epidemiology and the need for context-specific analysis.

The low number of AMR and MDR isolates in the 60–64 age group on the one hand and the high numbers of both AMR and MDR isolates among young children on the other emphasises the urgent need for targeted interventions to address the problem of AMR and MDR. The prevalence and distribution of AMR across different coagulase categories highlights public health and clinical importance of AMR. In view of this, there is a need for comprehensive strategies to address antimicrobial resistance.

The examination of antimicrobial agents reveals varying resistance patterns, with some antibiotics demonstrating high rates of resistance compared to others, which futher highlights the need for target interventions.

Finally, the analysis of predictors of MDR highlighted the source of the isolate and the season of isolation as important factors to be considered by clinicians when deciding on treatment options.

Overall, the findings of the present study underscore the complex nature of antimicrobial resistance and the need for multifaceted approaches to mitigate its impact on public health. Continued surveillance, prudent antibiotic use, and targeted interventions are essential to combatting antimicrobial resistance effective.

## Figures and Tables

**Figure 1 microorganisms-12-01862-f001:**
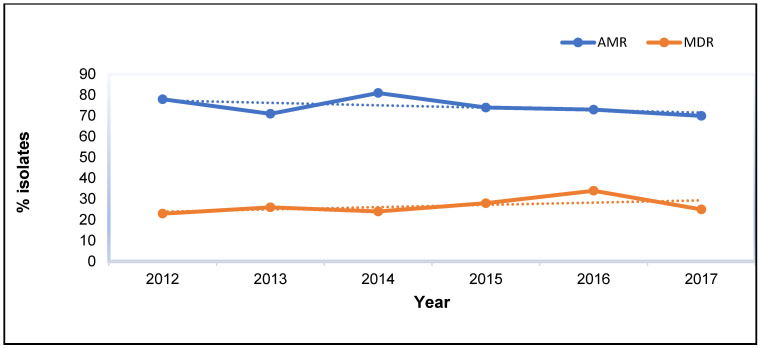
The annual temporal distribution of AMR and MDR *Staphylococcus* isolates from human samples at diagnostic laboratories in South Africa, 2012–2017.

**Table 1 microorganisms-12-01862-t001:** Distribution of antimicrobial resistance among *Staphylococcus* isolates from human samples by age and sex in South Africa, 2012–2017.

Variable	Total Isolates			AMR ^b^ Isolates			MDR^c^ Isolates		
	*n*	%	95% CI ^a^	*n*	%	95% CI ^a^	*n*	%	95% CI ^a^
Age groups	404,217			219,086	54.2		153,602		
≥65	22,517	5.6	0.544–0.576	11,392	5.2	0.504–0.536	7321	4.8	0.046–0.049
60–64	2438	0.6	0.054–0.066	7449	3.4	0.324–0.364	693	0.5	0.004–0.005
55–59	7049	1.7	0.164–0.176	3067	1.4	0.124–0.156	2070	1.3	0.013–0.014
50–54	11,770	2.9	0.284–0.304	8983	4.1	0.384–0.436	3159	2.1	0.020–0.022
45–49	14,962	3.7	0.366–0.374	5258	2.4	0.224–0.264	3516	2.3	0.022–0.024
40–44	19,620	4.9	0.486–0.494	11,612	5.3	0.504–0.564	4458	2.9	0.028–0.030
35–39	26,031	6.4	0.636–0.644	14,679	6.7	0.646–0.694	5047	3.3	0.032–0.034
30–34	33,685	8.3	0.826–0.834	17,308	7.9	0.766–0.812	9699	6.3	0.062–0.065
25–29	34,788	8.6	0.856–0.864	14,241	6.5	0.626–0.674	9198	6.0	0.058–0.062
20–24	25,897	6.4	0.636–0.644	12,926	5.9	0.564–0.616	6442	4.2	0.041–0.043
15–19	9610	2.4	0.236–0.244	9202	4.2	0.364–0.476	1957	1.3	0.012–0.014
10–14	6144	1.5	0.146–0.154	6573	3.0	0.276–0.324	3128	2.0	0.019–0.021
5–9	13,953	3.5	0.346–0.354	11,173	5.1	0.476–0.546	4756	3.1	0.030–0.032
0–4	117,566	29.1	0.289–0.291	81,500	37.2	0.364–0.386	74,378	48.4	0.481–0.488
Unknown	58,187	14.4	0.138–0.150	29,138	13.3	0.130–0.136	17,779	11.6	0.114–0.118
Sex	404,217			219,086			153,602		
Male	210,858	52.2	0.126–0.134	97,274	44.4	0.439–0.482	87,092	56.7	0.565–0.570
Female	181,270	44.8	0.443–0.453	91,359	41.7	0.413–0.420	60,673	39.5	0.392–0.397
Unknown	12,089	3.0	0.276–0.324	30,453	13.9	0.137–0.141	5837	3.8	0.037–0.039

^a^ 95% confidence interval; ^b^ AMR: antimicrobial resistant; ^c^ MDR: multidrug resistant.

**Table 2 microorganisms-12-01862-t002:** Distribution of antimicrobial-resistant *Staphylococcus* isolates by species of organisms from human samples in South Africa, 2012–2017.

Variable	Total Isolates			AMR ^b^ Isolates			MDR ^c^ Isolates		
	*n*	%	95% CI ^a^	*n*	%	95% CI ^a^	*n*	%	95% CI ^a^
Staph species	404,217			219,086			153,602		
CoPS ^d^	284,973	70.5		142,406	65.0		77,108	50.2	
*S. aureus*	270,421	66.9	0.665–0.672	131,233	59.9	0.598–0.600	69,474	90.1	0.896–0.906
*S. intermedius*	8084	2.0	0.188–0.212	5696	2.6	0.025–0.026	4168	6.0	0.562–0.638
*S. pseudintermedius*	6468	1.6	0.148–0.172	5477	2.5	0.024–0.025	163	3.9	0.362–0.418
CoNS ^e^	71,546	17.7		50,390	23		47,309	30.8	
*S. epidermidis*	45,272	11.2	0.109–0.144	23,004	10.5	0.836–0.846	34,678	73.3	0.729–0.737
*S. haemolyticus*	14,552	3.6	0.336–0.376	9640	4.4	0.434–0.448	5583	16.1	0.156–0.166
*S. hominis*	7276	1.8	0.168–0.192	11,393	5.3	0.522–0.538	402	7.2	0.682–0.758
*S. saprophyticus*	4446	1.1	0.108–0.112	6353	2.9	0.286–0.294	14	3.5	0.328–0.372
CoPS/CoNS ^f^	47,698	11.8		26,290	12.0		29,185	19.0	
Unspeciated *Staphylococcus*	36,784	9.1	0.902–0.920	15,117	6.9	0.686–0.694	25,945	88.9	0.884–0.894
*S. schleiferi*	5659	1.4	0.132–0.148	5258	2.4	0.234–0.244	1219	4.7	0.432–0.508
*S. hyicus*	5255	1.3	0.126–0.134	5915	2.7	0.264–0.274	78	6.4	0.602–0.678

^a^ 95% confidence interval; ^b^ AMR: antimicrobial resistant; ^c^ MDR: multidrug resistant; ^d^ CoPS: coagulase-positive *Staphylococcus*; ^e^ CoNS: coagulase-negative *Staphylococcus*; ^f^ CoPS/CoNS: coagulase-variable *Staphylococcus*.

**Table 3 microorganisms-12-01862-t003:** Distribution of antimicrobial-resistant isolates by antimicrobial classes among *Staphylococcus* isolates from human samples submitted to diagnostic laboratories in South Africa, 2012–2017.

	AMR ^b^ Isolates			MDR ^c^ Isolates		
	*n*	%	95% CI ^a^	*n*	%	95% CI ^a^
Beta-lactams:	136,862	33.9		106,917	36.9	
Ceftazidime	411	0.3	0.003–0.004	214	0.2	0.18–0.22
Amoxicillin	3832	2.8	0.027–0.029	1711	1.6	1.48–1.72
Ampicillin	3148	2.3	0.022–0.024	2352	2.2	2.08–2.32
Cloxacillin	96,147	70.3	0.701–0.706	79,332	74.2	73.96–74.44
Penicillin	19,982	14.6	0.144–0.148	14,755	13.7	13.68–13.92
Piperacillin	274	0.2	0.002–0.003	107	0.1	0.08–0.12
Cefazolin	274	0.2	0.001–0.003	107	0.1	0.08–0.12
Cefepime	411	0.3	0.002–0.003	214	0.2	0.18–0.22
Cefotaxime	821	0.6	0.005–0.006	428	0.4	0.38–0.42
Cefoxitin	10,128	7.4	0.073–0.075	6736	6.3	6.18–6.42
Cefuroxime	958	0.7	0.007–0.008	642	0.6	0.58–0.64
Ertapenem	67	0.0	0.000–0.000	100	0.1	0.00–0.02
Imipenem	274	0.2	0.001–0.002	214	0.2	0.18–0.22
Meropenem	137	0.1	0.001–0.001	107	0.1	0.08–0.12
Aminoglycosides:	53,081	13.1		40,494	14.1	
Gentamicin	51,655	97.3	0.992–0.993	37,457	92.5	0.991–0.994
Amikacin	657	1.2	0.001–0.003	1701	4.2	0.004-.0005
Streptomycin	330	0.6	0.000–0.001	486	1.2	0.000–0.001
Tobramycin	439	0.8	0.002–0.009	850	2.1	0.002–0.003
Sulphonamides:						
Co-trimoxazole	15,498	3.7	0.037–0.038	12,601	4.3	0.035–0.036
Tetracyclines:						
Tetracycline	46,311	12.1	0.120–0.122	31,721	10.9	0.108–0.110
Macrolides:						
Erythromycin	85,591	21.2	0.211–0.213	56,160	19.4	0.246–0.249
Lincosamides:						
Clindamycin	56,036	15.5	0.155–0.156	34,759	12.1	0.190–0.192
Phenicols:						
Chloramphenicol	10,490	2.5	0.024–0.025	7159	2.5	0.027–0.028
Polymyxins:						
Colistin	348	0.1	0.001–0.001	193	0.1	0.000–0.001

^a^ 95% confidence interval; ^b^ AMR: antimicrobial resistant; ^c^ MDR: multidrug resistant.

**Table 4 microorganisms-12-01862-t004:** Predictors of antimicrobial resistance among *Staphylococcus* spp. from human samples submitted to diagnostic laboratories in South Africa, 2012–2017.

Variable	AMR ^c^		
	AOR ^a^	95% CI ^b^	*p*-Value
Staph. species			
*S. aureus*	2.6	1.80–2.80	<0.001
*S. intermedius*	0.5	0.30–0.90	<0.001
*S. pseudintermedius*	Reference	Reference	Reference
Specimen source			
Skin	1.6	1.20–2.10	0.243
Urinary	0.8	0.60–1.60	0.384
Blood	0.4	0.20–2.20	0.067
Nasopharyngeal fluid	0.9	0.60–1.80	0.672
All other sites	Reference	Reference	Reference
Year	0.97	0.94–1.10	0.022

^a^ AOR: adjusted odds ratio; ^b^ 95% confidence interval; ^c^ AMR: antimicrobial resistance.

**Table 5 microorganisms-12-01862-t005:** Predictors of multidrug resistance among *Staphylococcus* spp. from different human samples submitted to diagnostic laboratories in South Africa, 2012–2017.

Variable	MDR ^c^		
	AOR ^a^	95% CI ^b^	*p*-Value
Specimen source			<0.001
Skin	2.7	0.80–11.60	0.246
Urinary	6.0	2.20–17.60	0.407
Blood	13.0	3.40–55.20	0.001
Nasopharyngeal fluid	2.4	0.80–7.80	0.265
Other sites (Reference)	1	-	-
Season			0.022
Autumn	0.9	0.80–1.30	0.152
Winter	1.8	1.20–2.70	0.045
Spring	1.6	0.40–4.60	0.355
Summer (Reference)	1	-	-

^a^ AOR: adjusted odds ratio; ^b^ 95% confidence interval; ^c^ MDR: multidrug resistant.

## Data Availability

The data supporting this study’s findings are available upon reasonable request and subject to specific conditions. For data access inquiries, including requests for collaboration or data-sharing agreements, please contact Thomas Papo, data analyst, at thomas.papo@nhls.ac.za. Requests will be evaluated on a case-by-case basis, considering the request’s nature, compliance with relevant regulations, and any associated agreements or protocols.

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
