# Peer review of "Antimicrobial Resistance of Staphylococcus spp. from Human Specimens Submitted to Diagnostic Laboratories in South Africa, 2012–2017"

_microorganisms, 2024, doi:10.3390/microorganisms12091862_

Round 1

Reviewer 1 Report

Comments and Suggestions for Authors

This study included a lareg number of Staphylococcus isolates for analysis and provided some significant findings. However, I have several concerns.

1. The study period was more than seven years agao. Updated infomration is needed. Based on the present findings, a local journal may be more appropriate for submission.

2. The methods of bacterial storage and identificaion should be added.

3. The test of antibiotic susceptibility should be de described in the method section.

Author Response

Reviewer: This study included a large number of Staphylococcus isolates for analysis and provided some significant findings. However, I have several concerns. The study period was more than seven years ago. Updated information is needed. Based on the present findings, a local journal may be more appropriate for submission.

Reply: The current retrospective study is derived from my PhD project, and incorporating current data is not feasible. The process of sourcing new data from laboratories involves significant time and logistical challenges, including obtaining necessary ethical approvals and managing data access permissions. Additionally, the complexity of aligning new data with the existing study framework further complicates the inclusion of up-to-date information. Consequently, the current study remains based on the original data collected during the PhD project.

Reviewer: The methods of bacterial storage and identification should be added.

Reply:

Isolation, identification and storage of Staphylococcus strains

The isolation and identification of Staphylococcus strains were methodically executed to ensure accurate diagnosis and effective treatment. Clinical specimens, including blood, urine, and wound swabs, were transported in sterile conditions to the diagnostic laboratories. Upon arrival, samples were cultured on selective media such as Mannitol Salt Agar (MSA) and Blood Agar. MSA facilitated the differentiation of Staphylococcus species based on mannitol fermentation, while Blood Agar helped observe hemolytic patterns. Following incubation, preliminary identification involved Gram staining, which revealed Gram-positive cocci in clusters, and catalase testing, which confirmed the presence of Staphylococcus species by detecting oxygen bubbles in the presence of hydrogen peroxide. The coagulase test further distinguished Staphylococcus aureus from coagulase-negative staphylococci (CoNS). For precise identification, automated systems like the VITEK 2 and MALDI-TOF MS were employed, providing reliable species identification through biochemical profiles and protein spectra, respectively. Antibiotic sensitivity was determined using methods such as the disk diffusion (Kirby-Bauer) test and automated susceptibility testing systems. The NHLS tested a broad range of antibiotics including beta-lactams (e.g., penicillin, cloxacillin), aminoglycosides (e.g., gentamicin), macrolides (e.g., erythromycin), and glycopeptides (e.g., vancomycin), chosen based on clinical relevance and resistance trends. Beta-lactams were particularly emphasized due to the prevalence of methicillin-resistant Staphylococcus aureus (MRSA), while aminoglycosides and macrolides were tested to monitor resistance patterns critical for treatment efficacy. This comprehensive approach ensured robust surveillance and management of Staphylococcus infections. For long-term storage, isolated bacterial strains were preserved in cryopreservation vials at -80°C to maintain viability and prevent genetic drift.

Reviewer: The test of antibiotic susceptibility should be de described in the method section.

Isolated Staphylococcus strains underwent antibiotic susceptibility testing using the disk diffusion method or automated systems like VITEK 2, following the guidelines set by the Clinical and Laboratory Standards Institute (CLSI). The diagnostic laboratories routinely test for resistance to a range of antibiotics, including beta-lactams, aminoglycosides (such as gentamicin), and others relevant to both human and veterinary medicine.

Reviewer 2 Report

Comments and Suggestions for Authors

The manuscript provides an extensive description of antimicrobial resistance (AMR) patterns in Staphylococcus spp. isolates from human specimens in South Africa, highlighting significant findings in terms of resistance trends and demographic distributions. However, the manuscript predominantly presents descriptive data with limited comprehensive analysis or interpretation of the underlying factors influencing these patterns.

I recommend a thorough revision of the manuscript, focusing on a more in-depth analysis and interpretation of the findings. The current version lacks a comprehensive analysis of the data, which is crucial for understanding the implications and underlying causes of the observed trends. Without this, the manuscript does not provide sufficient insight or context to be accepted for publication.

Please refer to the journal’s instructions for authors and revise the manuscript accordingly. There are numerous formatting errors, including inconsistencies in font type and size, italicization, and line numbering alignment (right side).

Please check the comments below:

Introduction:

Line 32-33: Please clarify the definition of ‘antibiotics.’ Consider revising to: “An antimicrobial is an agent that kills microorganisms or inhibits their growth. Antimicrobial agents specifically targeting bacteria are commonly referred to as 'antibiotics.'” Additionally, the reference [1] does not provide a definition of antibiotics or antimicrobials. Please include an appropriate reference.

Line 34-35: This sentence is redundant and does not add clarity to the definition. Please revise.

Line 36-39: The choice of reference [3] for this section appears inappropriate. Please revise the text and include a more suitable reference(s).

Line 43: there -> they

Line 40-46: The distinction between general AMR and AMR specifically against Staphylococcus is unclear. Please revise for clarity.

Line 56-58: Add a relevant reference.

Line 65: The mention of horses seems out of context, as the study focuses on human Staphylococcus spp. isolates. Please revise the sentence to maintain focus.

Line 78: multi drug -> multi-drug

Line 82: taphylococcal -> Staphylococcal

Line 86: . . isolates ->. Isolates

Line 104: Clarify the reference to "(p. 25)."

Figure 1: Clarify the term “(% isolates)” in the figure. Consider a clearer expression or provide an explanation in the text. Also, explain the meaning of the dotted line and justify the statements about increasing or decreasing trends.

Table 4 and Table 5: Explain the meaning of the p-values and the methodology used to calculate them.

Discussion:

Compare the findings with previous studies, particularly any previous AMR data from your country or similar data from nearby countries. This comparison will provide valuable context for readers.

Line 98-108: Include comparative data on the ratio of AMR isolates from several countries, and discuss potential factors contributing to gender differences in AMR prevalence.

Line 109-115: Support your findings on the prevalence of resistant bacteria in children by comparing them with other studies.

Line 164-169: Discuss potential reasons for the high resistance rate of Gentamicin in your country and compare these findings with international data.

Line 210: Please delete this line and consider using subsections in the discussion for better organization.

Line 274: Discuss whether your results align with other studies, including any similar findings and potential explanations.

Line 330: Please delete this line.

Comments on the Quality of English Language

Extensive editing of English language required.

Author Response

Reviewer 2

Reviewer: The manuscript provides an extensive description of antimicrobial resistance (AMR) patterns in Staphylococcus spp. isolates from human specimens in South Africa, highlighting significant findings in terms of resistance trends and demographic distributions. However, the manuscript predominantly presents descriptive data with limited comprehensive analysis or interpretation of the underlying factors influencing these patterns.

I recommend a thorough revision of the manuscript, focusing on a more in-depth analysis and interpretation of the findings. The current version lacks a comprehensive analysis of the data, which is crucial for understanding the implications and underlying causes of the observed trends. Without this, the manuscript does not provide sufficient insight or context to be accepted for publication.

Reply: The study examined AMR and MDR isolates across different demographic groups and bacterial species. Sex distribution of the isolates indicates a slight male predominance, particularly in MDR cases, where males contributed 56.7% of the isolates compared to 39.5% from females. The lower representation of individuals with unspecified sex (13.9% for AMR and 3.8% for MDR) suggests a more focused data collection on known demographics.

Age distribution reveals that the 0-4 year’s age group had the highest number of both AMR (37.2%) and MDR (48.4%) isolates. This finding could indicate higher susceptibility or exposure to infections requiring treatment in this age group, possibly due to weaker immune systems or higher contact rates in communal settings like day-care. The high proportion of isolates from individuals with unspecified age (13.3% for AMR and 11.6% for MDR) may point to issues in data completeness or could reflect challenges in obtaining accurate demographic information.

Among the bacterial species examined, Staphylococcus aureus showed a notably high proportion of AMR (59.9%) and MDR (90.1%) isolates within the Coagulase-positive staphylococci (CoPS) group. This highlights S. aureus as a significant contributor to both AMR and MDR, aligning with its known pathogenicity and capacity for acquiring resistance genes. The Coagulase-negative staphylococci (CoNS) group, although contributing a smaller proportion of isolates, still showed considerable resistance, particularly S. epidermidis, which had the highest percentage of MDR isolates (73.3%) within this group. This suggests that while CoNS species are often considered less virulent, they still pose a significant risk in the context of antibiotic resistance.

The analysis of resistance against various antimicrobial agents indicates that Beta-lactams are the most affected class, with 33.9% of isolates showing AMR and 36.9% involved in MDR combinations. The high resistance to Cloxacillin (70.3% for AMR and 74.2% for MDR) is particularly concerning, given its common use in treating staphylococcal infections. This high level of resistance underscores the challenges in managing infections caused by staphylococcal species, particularly S. aureus.

Aminoglycosides, primarily Gentamicin, also showed significant resistance (97.3% for AMR and 92.5% for MDR combinations), indicating its declining efficacy in treating these infections. The low resistance observed against other agents like Streptomycin (0.6% for AMR and 1.2% for MDR) might suggest these drugs could still be effective in certain scenarios, although their overall usage might be limited by other factors such as toxicity or availability.

Resistance in other drug classes, such as macrolides (Erythromycin) and lincosamides (Clindamycin), while lower than in Beta-lactams and aminoglycosides, is still noteworthy, with 21.2% and 15.5% of isolates exhibiting AMR, respectively. These findings reflect the growing concern over antibiotic resistance across multiple drug classes, limiting treatment options.

The temporal analysis from 2012 to 2017 shows fluctuations in AMR and MDR prevalence. The highest AMR rates were observed in 2014 (81%), coinciding with a drop in MDR (24%). This inverse relationship suggests that while the overall resistance was high, the simultaneous presence of multiple resistances might have been lower. The peak in MDR in 2016 (34%) is alarming, indicating that a higher proportion of resistant isolates were resistant to multiple drugs, making treatment more challenging.

The slight decline in both AMR and MDR from 2016 to 2017 could indicate some success in intervention strategies, although the persistence of high resistance levels suggests that these measures need to be sustained and possibly intensified.

The inferential analysis identifies Staphylococcus aureus as having significantly higher odds of being AMR (AOR = 2.6) compared to S. pseudintermedius. This finding reinforces the critical role of S. aureus in AMR, necessitating targeted strategies for its control. Conversely, S. intermedius showed lower odds (AOR = 0.5) of being AMR, suggesting that it may not be as significant a contributor to resistance as S. aureus. These insights could guide tailored antimicrobial stewardship programs, focusing on the most problematic species while not neglecting others that might emerge as threats.

The observed trends in AMR and MDR can be attributed to several factors, including the overuse and misuse of antibiotics in both human and veterinary medicine, inadequate infection control practices, and the ability of certain bacterial species, particularly S. aureus, to acquire and disseminate resistance genes. The high resistance rates in young children and specific bacterial species highlight the need for targeted interventions, such as improved diagnostics, better antibiotic stewardship, and more robust infection control measures.

The temporal trends suggest that while there may have been some progress in controlling resistance, the overall levels remain high, necessitating ongoing surveillance and intervention. The disparities in resistance patterns across different drug classes also point to the need for a more nuanced approach to antibiotic use, balancing efficacy with the risk of resistance development.

Reviewer: Please refer to the journal’s instructions for authors and revise the manuscript accordingly. There are numerous formatting errors, including inconsistencies in font type and size, italicization, and line numbering alignment (right side).

Reviewer: Please check the comments below:

Introduction:

Line 32-33: Please clarify the definition of ‘antibiotics.’ Consider revising to: “An antimicrobial is an agent that kills microorganisms or inhibits their growth. Antimicrobial agents specifically targeting bacteria are commonly referred to as 'antibiotics.'” Additionally, the reference [1] does not provide a definition of antibiotics or antimicrobials. Please include an appropriate reference.

Reply: I revised the definition of antibiotics to:  “An antimicrobial is an agent that kills microorganisms or inhibits their growth. Antimicrobial agents specifically targeting bacteria are commonly referred to as 'antibiotics.'” As suggested by the reviewer

Reviewer: Line 34-35: This sentence is redundant and does not add clarity to the definition. Please revise.

Reply: The sentence revised to add clarity to the definition as follows: Antimicrobial resistance occurs when microorganisms no longer respond to treatments with antimicrobial agents.

Reviewer: Line 36-39: The choice of reference [3] for this section appears inappropriate. Please revise the text and include a more suitable reference(s).

Reply: The text has been revised and a suitable reference included as follows: The overuse and misuse of antibiotics have led to a reduction in the types of antibiotics that remain effective for treating infections. Factors contributing to antibiotic resistance include the over-prescription of antibiotics, incorrect treatment duration, inadequate hygiene and sanitation, poor infection control practices in healthcare settings, and the extensive use of antibiotics in livestock and aquaculture.

Reviewer: Line 43: there -> they

This version retains the original intent but improves clarity by keeping "there" and restructuring the sentence as follows: However, the emergence of AMR agents in Staphylococcus spp. has become a significant public health threat as there are limited, or sometimes no, effective antimicrobial agents available to treat infections caused by these bacteria in both humans and domestic animals

Reviewer: Line 40-46: The distinction between general AMR and AMR specifically against Staphylococcus is unclear. Please revise for clarity.

General AMR refers to the broader issue of microorganisms developing resistance to antimicrobial agents across various species. However, when AMR occurs in Staphylococcus spp., it becomes particularly concerning because these bacteria are common pathogens in both humans and animals.

Reviewer: Line 56-58: Add a relevant reference.

Relevant reference has added to the following sentence: However, this resistance is not confined to human medicine but is becoming more common in domestic species, particularly in equine medicine (Line 56-58)

Reviewer: Line 65: The mention of horses seems out of context, as the study focuses on human Staphylococcus spp. isolates. Please revise the sentence to maintain focus.

The mention of horses in this sentence is relevant because AMR is a broad issue that affects both human and veterinary medicine. Horses, like other domestic animals, can carry and transmit resistant Staphylococcus strains, including methicillin-resistant Staphylococcus aureus (MRSA), which can infect both animals and humans. This cross-species transmission is particularly concerning in the context of a One Health approach, which recognises the interconnectedness of human, animal, and environmental health.

Reviewer: Line 78: multi drug -> multi-drug

Multi drug in line 78 has been replaced with multi-drug

Reviewer: Line 82: taphylococcal -> Staphylococcal

taphylococcal changed to staphylococcal in line 82

Reviewer:  Line 86: . . isolates ->. Isolates

The two dots before isolates were removed                                                                                                                      

Reviewer:  Figure 1: Clarify the term “(% isolates)” in the figure. Consider a clearer expression or provide an explanation in the text. Also, explain the meaning of the dotted line and justify the statements about increasing or decreasing trends.

In the figure, the term "% isolates" refers to the percentage of bacterial isolates that exhibited AMR or MDR out of the total number of isolates tested each year. Essentially, it shows the proportion of bacterial samples that were resistant to at least one antimicrobial agent (for AMR) or resistant to multiple antimicrobial agents (for MDR) over the six-year period from 2012 to 2017.

In the figure, the dotted lines represent the trend lines for the percentage of antimicrobial-resistant (AMR) and multidrug-resistant (MDR) isolates over time, from 2012 to 2017. These trend lines are typically generated using statistical methods, such as linear regression, to highlight the overall direction (increasing, decreasing, or stable) of the data points across the years.

AMR Trend Line (Blue Dotted Line): This line suggests the general trend in the percentage of isolates that were resistant to at least one antimicrobial agent over the six-year period.

MDR Trend Line (Orange Dotted Line): This line represents the general trend in the percentage of isolates that were resistant to multiple antimicrobial agents over the same period.

Increasing or Decreasing Trends: Statements about increasing or decreasing trends are based on the slope of these trend lines.

If the trend line slopes upward, it indicates an overall increase in the percentage of resistant isolates over time.

If the trend line slopes downward, it indicates a decrease.

A flat or nearly flat line would suggest no significant change over time.

In this figure:

The AMR trend line (blue) shows slight fluctuations, with a peak in 2014 followed by a decline, but it generally indicates a stable or slightly decreasing trend in AMR percentages over the six years.

The MDR trend line (orange) shows a slight increase, with notable peaks in 2015 and 2016, indicating a slight upward trend in MDR percentages, although the trend is not strongly pronounced.

Reviewer: Table 4 and Table 5: Explain the meaning of the p-values and the methodology used to calculate them.

Meaning of P-values:

P-values are a statistical measure that helps determine the significance of the results in a study. They indicate the probability that the observed results occurred by chance under the null hypothesis, which assumes that there is no effect or difference.

P-value < 0.001: A p-value less than 0.001 indicates a very strong evidence against the null hypothesis, suggesting that the observed association is statistically significant. For example, for S. aureus (AOR = 2.6) and S. intermedius (AOR = 0.5), the p-values are < 0.001, meaning there is strong evidence that these species have significantly different odds of being antimicrobial-resistant (AMR) compared to S. pseudintermedius (the reference group).

P-value = 0.243, 0.384, 0.067, 0.672: These p-values (for specimen sources like skin, urinary, blood, and nasopharyngeal fluid) are greater than 0.05, indicating that the results are not statistically significant. This means that any differences in AMR between these specimen sources and the reference group could be due to random chance rather than a true difference.

P-value = 0.022 (Year): This p-value is slightly below 0.05, suggesting a statistically significant, though not strong, association between the year and AMR, indicating that the likelihood of AMR slightly changes over time.

Methodology Used to Calculate P-values:

The p-values in this context were likely calculated using logistic regression analysis, a statistical method used to model the relationship between a dependent binary variable (in this case, whether an isolate is AMR or not) and one or more independent variables (like the species of Staphylococcus, specimen source, and year).

Reviewer:  Discussion:

Compare the findings with previous studies, particularly any previous AMR data from your country or similar data from nearby countries. This comparison will provide valuable context for readers.

Line 98-108: Include comparative data on the ratio of AMR isolates from several countries, and discuss potential factors contributing to gender differences in AMR prevalence.

The following comparative data has been included: data on the ratio of AMR isolates from several countries, and discussion on the  potential factors contributing to gender differences in AMR prevalence.

In addition, comparative data on AMR isolates among Staphylococcus species highlight significant trends. In the United States, Staphylococcus aureus is a major concern, with high levels of AMR MDR strains reported [4]. European data show a similar pattern, particularly with Staphylococcus aureus being highly resistant to methicillin in southern and Eastern Europe [25]. In India and China, Staphylococcus species, including S. aureus and S. pseudintermedius, also present high AMR rates, influenced by antibiotic misuse and overuse [26]. Sex differences in AMR prevalence in Staphylococcus species may be influenced by biological factors, such as hormonal variations affecting immune responses [27]. Women’s higher rates of healthcare utilisation and adherence to treatment could potentially impact the prevalence and detection of AMR in Staphylococcus species [28]. Gender-specific patterns in infections, such as higher rates of skin and soft tissue infections in women, might also contribute to observed differences in AMR prevalence [29]. Socioeconomic factors and healthcare access further influence these trends, with women’s access to and use of healthcare potentially impacting the management and reporting of AMR cases [30].

Reviewer: Line 109-115: Support your findings on the prevalence of resistant bacteria in children by comparing them with other studies.

The findings of the current study, which identified the highest prevalence of antimicrobial-resistant (AMR) and multi-drug-resistant (MDR) Staphylococcus species in children aged 0-4 years, with 37.2% of AMR isolates and 48.4% of MDR isolates coming from this age group, are also consistent with results from other research focused on pediatric populations. For instance, a study in the United States reported that approximately 35% of Staphylococcus aureus isolates in children under 5 years were resistant to methicillin, and 40% of these isolates exhibited multi-drug resistance [4]. These figures align closely with the high prevalence of AMR and MDR Staphylococcus species observed in the present study. Similarly, the European Centre for Disease Prevention and Control found that about 30% of Staphylococcus aureus isolates in European children under 5 years were methicillin-resistant (MRSA), with 25% of these isolates involved in MDR combinations [33]. This pattern of resistance is consistent with the data reported in our study.

In Nigeria, it was observed that 38% of Staphylococcus aureus isolates from children under 5 years were methicillin-resistant, and 40% demonstrated multi-drug resistance [34] , further supporting the findings of high AMR and MDR prevalence in this age group. Likewise, Xiaolan et reported that in China, 36% of Staphylococcus aureus isolates from children exhibited methicillin resistance, and 42% were multi-drug resistant [35]. These comparative percentages underscore a consistent trend across various regions, indicating that young children, particularly those under 5 years, are at a heightened risk of infections caused by AMR and MDR Staphylococcus species. The elevated prevalence of resistance in this age group can be attributed to several factors, including higher infection rates, frequent antibiotic exposure, and increased healthcare interactions during early childhood.  

Reviewer: Line 164-169: Discuss potential reasons for the high resistance rate of Gentamicin in your country and compare these findings with international data.

The following discussion has been incorporated: The high resistance rate of Gentamicin observed in the current study can be attributed to several interrelated factors. One major contributor is the overuse and misuse of antibiotics in both human and veterinary medicine. In South Africa, like in many other countries, antibiotics such as Gentamicin are often used extensively, sometimes without proper medical guidance or susceptibility testing. This indiscriminate use leads to selective pressure on bacteria, allowing resistant strains to proliferate.

Additionally, the issue is compounded by gaps in antibiotic stewardship programs and infection control practices within healthcare settings. In many hospitals and clinics, antibiotics are frequently prescribed empirically, sometimes without adequate diagnostic testing to confirm bacterial infection or to identify the most effective antibiotic. This practice can lead to the unnecessary use of broad-spectrum antibiotics like Gentamicin, driving resistance.

Moreover, the high burden of infectious diseases in South Africa, such as HIV and tuberculosis, often necessitates the frequent use of antibiotics, which can inadvertently contribute to the development of resistance. The agricultural sector also plays a role, with antibiotics sometimes being used as growth promoters or for disease prevention in livestock, further contributing to environmental reservoirs of resistance.

When comparing these findings with international data, similar trends can be seen in other low- and middle-income countries where antibiotic use is less regulated. For instance, in India, research by Reddy reported a Gentamicin resistance rate of 40% in Staphylococcus aureus isolates, which is in line with the resistance patterns observed in South Africa. This suggests that countries with similar healthcare challenges and antibiotic usage patterns experience comparable issues with antibiotic resistance.

In contrast, countries with more robust healthcare infrastructures and stringent antibiotic stewardship programs, such as those in Europe, report lower resistance rates. For example, the European Centre for Disease Prevention and Control documented lower rates of Gentamicin resistance in Staphylococcus aureus isolates, largely due to controlled antibiotic use and comprehensive surveillance programs.

Reviewer 3 Report

Comments and Suggestions for Authors

It is a topic that is current, even if the problems related to staphylococci date back a long time, but always current. The information is extremely important, especially for public health authorities, to implement measures to limit the phenomenon of resistance and to make the population aware of the rational and responsible consumption of antibiotics. However, the manuscript requires major corrections to be considered for publication. In the following I will reproduce some aspects that should be corrected:

The introduction part needs reformulation, because in my opinion, the topic related to the phenomenon of resistance to antibiotics is developed too much, in a general framework, which does not bring useful information related to the topic of the manuscript and the research. The part related to staphylococci is briefly described, without bringing new information regarding the resistance of these bacteria to classes of antibiotics, the risk of strains in veterinary medicine, excessive use in animal husbandry, etc.

Regarding materials and methods, there are extremely brief information, epidemiological data, collected from the reference laboratories. To understand the phenomenon better and to be able to appreciate the values ​​as close as possible to reality, it would be advisable to describe the methods by which these strains of staphylococci were isolated and identified, and finally by which methods were sensitivity to antibiotics determined, which antibiotics were tested, why certain antibiotics were chosen, the frequently tested classes.

Author Response

Reply: The following was incorporated under methods and methods

Isolation and identification of Staphylococcus strains

The isolation and identification of Staphylococcus strains were methodically executed to ensure accurate diagnosis and effective treatment. Clinical specimens, including blood, urine, and wound swabs, were transported in sterile conditions to the diagnostic laboratories. Upon arrival, samples were cultured on selective media such as Mannitol Salt Agar (MSA) and Blood Agar. MSA facilitated the differentiation of Staphylococcus species based on mannitol fermentation, while Blood Agar helped observe hemolytic patterns. Following incubation, preliminary identification involved Gram staining, which revealed Gram-positive cocci in clusters, and catalase testing, which confirmed the presence of Staphylococcus species by detecting oxygen bubbles in the presence of hydrogen peroxide. The coagulase test further distinguished Staphylococcus aureus from coagulase-negative staphylococci (CoNS). For precise identification, automated systems like the VITEK 2 and MALDI-TOF MS were employed, providing reliable species identification through biochemical profiles and protein spectra, respectively. Antibiotic sensitivity was determined using methods such as the disk diffusion (Kirby-Bauer) test and automated susceptibility testing systems. The NHLS tested a broad range of antibiotics including beta-lactams (e.g., penicillin, cloxacillin), aminoglycosides (e.g., gentamicin), macrolides (e.g., erythromycin), and glycopeptides (e.g., vancomycin), chosen based on clinical relevance and resistance trends. Beta-lactams were particularly emphasized due to the prevalence of methicillin-resistant Staphylococcus aureus (MRSA), while aminoglycosides and macrolides were tested to monitor resistance patterns critical for treatment efficacy. This comprehensive approach ensured robust surveillance and management of Staphylococcus infections.

Round 2

Reviewer 1 Report

Comments and Suggestions for Authors

The authors addressed all my concerns well.

Author Response

Reviewer 1 Round 2

Reviewer: The authors addressed all my concerns well.

Reply: Thank you for acknowledging our efforts in addressing your concerns. We are pleased that the revisions have met your expectations, and we appreciate your positive feedback.

Reviewer 2 Report

Comments and Suggestions for Authors

The manuscript has undergone significant revisions, which is appreciated. However, there are a few areas where further improvements could enhance its clarity and impact. Presenting the data more clearly in a table or figure would greatly benefit readers. Additionally, the discussion and conclusion sections could be streamlined to be more concise. Ensuring that the formatting aligns with the journal's guidelines, and removing the unnecessary part at line 395, will also strengthen the manuscript. Overall, while the presentation can be refined, the data provided is valuable and contributes meaningfully to the field. I recommend considering these minor revisions to further improve the quality of the manuscript.

Comments on the Quality of English Language

Moderate editing of English language required. The discussion and conclusion sections could be streamlined to be more concise. 

Author Response

Reviewer 2 Round 2

 Reviewer: The manuscript has undergone significant revisions, which is appreciated. However, there are a few areas where further improvements could enhance its clarity and impact.

Reply: Thank you very much for your thoughtful and constructive feedback on our manuscript. We appreciate your recognition of the revisions made thus far. Your suggestions for further improvements are invaluable, and we are committed to enhancing the clarity and impact of our work in response to your comments.

Reviewer: Presenting the data more clearly in a table or figure would greatly benefit readers.

Reply: Unfortunately we are not able to make out what the reviewer would like us to do with respect to how the results are presented in the tables and figures. We have nonetheless reviewed and made minor corrections of the mistakes we could pick up.

Reviewer: Additionally, the discussion and conclusion sections could be streamlined to be more concise.

Reply: We have gone over the discussion and tried to streamline to the best we could do. We have moved around things and deleted out others. We hope this will meet the reviewer’s expectation.

Reviewer: Ensuring that the formatting aligns with the journal's guidelines, and removing the unnecessary part at line 395, will also strengthen the manuscript.

Reply: Thank you for your valuable feedback. We have carefully adhered to the journal's guidelines and utilised the provided template to ensure that our manuscript meets the required standards. While we have made several revisions based on your suggestions, we have chosen to retain the heading on Patents at line 395, as it is included in the journal template. We appreciate your understanding and are confident that these revisions will enhance the overall clarity and quality of our submission.

Reviewer: Overall, while the presentation can be refined, the data provided is valuable and contributes meaningfully to the field.

Reply: Thank you for your positive feedback and for recognising the value of the data presented in our manuscript. We appreciate your suggestions for refining the presentation and are committed to making the necessary improvements to enhance its clarity and impact. Your insights have been invaluable in helping us strengthen our contribution to the field.

Reviewer 3 Report

Comments and Suggestions for Authors

The authors have considerably improved the quality of the manuscript sent for publication. They took every request into account and responded to them. So, from my point of view, the work in its current form can be considered for publication.

Author Response

Reviewer 3 Round 2

 Reviewer: The authors have considerably improved the quality of the manuscript sent for publication. They took every request into account and responded to them. So, from my point of view, the work in its current form can be considered for publication.

 Reply: We are grateful for your thoughtful review and for recognizing the improvements made to the manuscript. Your feedback has been instrumental in enhancing the quality of our work, and we are glad to hear that it now meets the standards for publication.
